# Local Characteristics Related to SARS-CoV-2 Transmissions in the Seoul Metropolitan Area, South Korea

**DOI:** 10.3390/ijerph182312595

**Published:** 2021-11-29

**Authors:** Changmin Im, Youngho Kim

**Affiliations:** 1Department of Geography, Korea University, 145 Anam-ro, Seoul 02841, Korea; changminim@korea.ac.kr; 2Department of Geography & Geography Education, Korea University, 145 Anam-ro, Seoul 02841, Korea

**Keywords:** COVID-19, young adults, Christians, subway commuters, geographically weighted lasso

## Abstract

The Seoul metropolitan area is one of the most populated metropolitan areas in the world; hence, Seoul’s COVID-19 cases are highly concentrated. This study identified local demographic and socio-economic characteristics that affected SARS-CoV-2 transmission to provide locally targeted intervention policies. For the effective control of outbreaks, locally targeted intervention policies are required since the SARS-CoV-2 transmission process is heterogeneous over space. To identify the local COVID-19 characteristics, this study applied the geographically weighted lasso (GWL). GWL provides local regression coefficients, which were used to account for the spatial heterogeneity of SARS-CoV-2 outbreaks. In particular, the GWL pinpoints statistically significant regions with specific local characteristics. The applied explanatory variables involving demographic and socio-economic characteristics that were associated with higher SARS-CoV-2 transmission in the Seoul metropolitan area were as follows: young adults (19~34 years), older population, Christian population, foreign-born population, low-income households, and subway commuters. The COVID-19 case data were classified into three periods: the first period (from January 2020 to July 2021), the second period (from August to November 2020), and the third period (from December 2020 to February 2021), and the GWL was fitted for the entire period (from January 2020 to February 2021). The result showed that young adults, the Christian population, and subway commuters were the most significant local characteristics that influenced SARS-CoV-2 transmissions in the Seoul metropolitan area.

## 1. Introduction 

The Seoul metropolitan area is one of the most vulnerable metropolitan areas for SARS-CoV-2 transmission in the world. Since SARS-CoV-2 spreads more easily in crowded places [1], outbreaks are highly concentrated in metropolitan areas [2]. The Seoul metropolitan area’s population density is the 10th highest among 670 metropolitan areas in Organization for Economic Cooperation and Development (OECD) countries [3]. Consequently, Korean COVID-19 cases are concentrated in the Seoul metropolitan area. All the confirmed cases in the Seoul metropolitan area represent about 64.9% of the total cases in the country (53,610 confirmed cases reported as of 28 February 2021).

To prevent SARS-CoV-2 transmission in the Seoul metropolitan area, the Korean government enacted intervention policies, such as wearing facial masks and nationwide social distancing, from 29 February 2020 [4]. Despite stringent intervention policies that lasted for one and a half years, the Seoul metropolitan area has shown the highest incidence rate in Korea. The area reported around 400 cases per day, while all other metropolitan areas confirmed up to 100 cases on average (as of 10 June 2021).

The SARS-CoV-2 transmission process is heterogeneous over space [5,6,7]. Local demographic and socio-economic characteristics affect the SARS-CoV-2 transmission process [8]. This study found the cause of failure in COVID-19 control from uniform and standardized intervention policies in the Seoul metropolitan area, which ignored detailed local characteristics. Naturally, locally targeted intervention policies are required for advanced and effective COVID-19 control [9,10]. This study argued that the locally targeted intervention policies enable the control of SARS-CoV-2 transmission at the regional level [8,9].

Recent COVID-19 analyses identified the local characteristics that are related to COVID-19 to allow for local intervention policies [11,12,13,14,15,16]. In particular, local COVID-19 studies implement local spatial regression statistics [14,15,16,17,18,19,20,21,22]. Local spatial regression models provide spatially varying coefficients by applying region-specific information [23,24]. Previous research, which adopted global spatial models, only provided generalized conclusions that did not take into account region-specific information [25,26,27,28,29]. Such a global model, in turn, would mislead public health policy makers [30]. Compared to the global model, the local coefficients are more appropriate for understanding the characteristics of COVID-19 transmission at the regional level.

This study applied the geographically weighted lasso (hereafter GWL) and accounted for local characteristics of the transmission at the regional level [24]. The GWL has the strength of providing locally targeted results with reliable statistical inferences about the local characteristics of COVID-19 in the Seoul metropolitan area. Compared with geographically weighted regression (hereafter GWR), which is the most widely used local spatial regression, the GWL incorporates the lasso (the least absolute shrinkage and selection operator) into local regressions, accounting for multicollinearity issues [24]. In addition, the lasso shrinks less significant coefficients to zero in regressions while maintaining the interpretability of the independent variables [31].

This study is distinguished by the fact that it applied the GWL to account for SARS-CoV-2 transmission. First, this study provided statistically more reliable results and interpretation since the GWL is free from multicollinearity issues. Most of the research that adopted GWR as a local spatial regression tool is affected by multicollinearity [14,15,16,17,18,19,20,21], which, in turn, leads to biased results and interpretation [32]. Second, the GWL results provided mapping that was suitable for locally targeted intervention policy regarding COVID-19. The GWL specified regions with more significant relationships of each variable since the lasso shrunk the less significant variables’ coefficients to zero by region [24,31]. Given that the variables used in the GWL accounted for changes in COVID-19 incidence over time in the Seoul metropolitan area, the GWL map results would help to practically advance locally targeted intervention.

After applying the GWL, the results showed that SARS-CoV-2 transmissions were significantly influenced by population groups with higher mobility (young adults, the Christian population, and subway commuters) from January 2020 to February 2021. In particular, subway commuters played an important role in transmitting SARS-CoV-2 all over the Seoul metropolitan area. In addition, socially vulnerable groups (the older population, foreign-born population, and low-income households) were more susceptible to SARS-CoV-2 transmission by region.

This study identified the most significant local characteristics related to SARS-CoV-2 transmission and specified statistically significant regions for the applied local variables. This study strongly suggested that the Korean government should reduce the crowdedness in subways, at least during rush hours, for more effective SARS-CoV-2 transmission control. In addition, local intervention policies should target regions with larger numbers of socially vulnerable groups, focusing on migrant workers, medical scarcity, and subway overcapacity. This study helps to advance intervention policies for COVID-19 in Seoul metropolitan area.

## 2. Data and Methods

### 2.1. Data

#### 2.1.1. Study Data

The study area was the Seoul metropolitan area, which consists of three principal administrative regions: Seoul, Gyeonggi-do, and Incheon (Figure 1). The Seoul metropolitan area comprises about 9.9 million people living in Seoul, 13.8 million in Gyeonggi-do, and 3 million in Incheon. The three principal administrative regions have 77 sub-administrative areal units (city and county level): 25 counties in Seoul, 42 cities and counties in Gyeonggi-do, and 10 counties in Incheon. These 77 cities and counties were used as study units for the regression models.

#### 2.1.2. COVID-19 Cases

This study used 14 months’ worth of COVID-19 data from each of the three administrative areas considered (https://data.seoul.go.kr; https://www.gidcc.or.kr; and https://www.incheon.go.kr—accessed on 13 March 2021). From January 2020 to February 2021, a total of 53,610 positive COVID-19 cases were reported (Figure 2). As shown in Figure 2, this study grouped the COVID-19 cases into three periods: (a) the first period (from January 2020 to July 2021), (b) the second period (from August to November 2020), and (c) the third period (from December 2020 to February 2021). This classification corresponds to changes in the Korean COVID-19 intervention policy levels. The levels presented in Figure 2 correspond to the social distancing levels as intervention policies in the Seoul metropolitan area. Up to February 2021, social distancing levels were divided into five levels: 1, 1.5, 2, 2.5, and 3 [33]. As the levels rose, restrictions on public gathering and mobility were reinforced.

In the first period, the average number of daily COVID-19 cases was about 16 cases. During the same period, mass infection cases broke out at nightclubs in Itaewon, Seoul. Mass infection cases occurred at Itaewon and Yongsan-gu from 30 April 2020 to 20 May 2020; Itaewon is one of the most representative nightlife places in the Seoul metropolitan area and more than 200 infection cases have been traced to nightclubs in Itaewon. In the second period, another mass infection occurred in Christian churches in the Seoul metropolitan area. In particular, Sarang Jeil Church in Seoul showed the highest number, with 1173 reported cases [34]. In the third period, COVID-19 cases exploded in the Seoul metropolitan area (Figure 2).

#### 2.1.3. Demographic and Socio-Economic Variables

This study used demographic and socio-economic variables to investigate the relationships with COVID-19 cases in the Seoul metropolitan area. The selected variables were identified as relevant risk factors for SARS-CoV-2 outbreaks. The descriptive statistics and references for the variables used in this study are presented in Table 1.
Young adults (aged 19 to 34 years)

Young adults aged 19 to 34 years were identified as one of the main spreaders of SARS-CoV-2. For example, young adults in the United States were responsible for more than 70% of the spread of SARS-CoV-2 in 2020 [36]. Most young people tend to have an inaccurate perception of infection risks since severe COVID-19 cases are rare in young [37]. However, young adults are also susceptible to infection due to frequent exposure to areas with high COVID-19 risks [37].
Older population (aged 65 years and above)

The older population is the most vulnerable population group to the COVID-19 pandemic [41]. The population aged over 60 years accounts for about 25% of total COVID-19 cases in Korea [61]. Facing more infection risks than the young, the elderly are more inclined to comply with intervention policies, such as social distancing and quarantine [38,39].
Christian population

Religion serves as a hub of SARS-CoV-2 transmission [45]. Religious practices, such as church attendance and worship, are directly linked to the spread of the virus. On 8 July 2020, the Korean government banned religious events and small gatherings to prevent SARS-CoV-2 transmission [62]. However, some Christian churches did not comply with the government guidelines [42]. As a result, COVID-19 cases from the Christian churches accounted for the highest proportion of mass infection cases in Korea [63].
Foreign-born population

The foreign-born population plays a role in SARS-CoV-2 transmission, importing the disease and initiating outbreaks [48,49]. In addition, as a socially vulnerable population, the foreign-born population is more susceptible to infectious disease [46,47]. As of March 2021, COVID-19 cases from the foreign-born population accounted for about 10 percent of the total cases in Korea [64]. The foreign-born population data refer to registered non-Korean nationals living in the Seoul metropolitan area, comprising about 770,000 people [65].
Low-income households

Low-income households are populations that are exposed to high SARS-CoV-2 infection risks [51,52]. Low economic status is a risk factor for infectious disease transmission [27,28,53,54,66]. Naturally, in Korea, low economic status is related to an increased risk of SARS-CoV-2 infection [50]. The low-income households that were used in this study refer to populations that were supported by Korean government funding.
Subway commuters

Commuters using subways act as SARS-CoV-2 transmission vectors [56,57]. Public transportation had significant impacts on the SARS-CoV-2 spread [58,59,60]. The New York case exemplifies the impact of public transportation on SARS-CoV-2 transmission, showing that the incidence rate decreased rapidly during the subway shutdown [56].

## 2.2. Method

### 2.2.1. Poisson Regression Model

This study used COVID-19 case data as a dependent variable that followed a Poisson distribution. Given that Poisson regression is a generalized linear model with a Poisson distribution error structure and the natural log (ln) link function [67], a Poisson regression model can be described as
$$log\left(\mu \right)=\beta_{0}+\beta_{1}X_{1}+\beta_{2}X_{2}+\cdots +\beta_{k}X_{k},$$
where μ is the expected case on the dependent variable given the independent variables $X_{1},X_{2},\cdots ,X_{k}$. In addition, $\beta_{0}$ is the intercept, and $\beta_{k}$ is the regression coefficient for the independent variable $X_{k}$. The equation implies that the logarithm of the mean μ is linearly related to the values of independent variables $X_{1},X_{2},\cdots ,X_{k}$. The equation above can be written as an exponential relationship between the mean and the independent variables:$$\mu =exp\left(\beta_{0}+\beta_{1}X_{1}+\beta_{2}X_{2}+\cdots +\beta_{k}X_{k}\right)=exp\left(\beta_{0}\right)\times exp\left(\beta_{1}\right)X_{1}\times \cdots \times exp\left(\beta_{k}\right)X_{k}$$

From the equation above, $exp\left(\beta_{k}\right)$ is interpreted as a multiplicative effect on the expected case for a one-unit increase in $X_{k}$ when holding other variables constant [68].

### 2.2.2. Geographically Weighted Lasso (GWL)

A GWL is a regression model that deals with spatial heterogeneity by adding a lasso (least absolute shrinkage and selection operator) constraint to GWR in parameter estimation [24]. GWR is a regression modeling technique that is used to fit a regression to each location based on the neighbors within a specific bandwidth [69]. For each observation location, *i* = 1,…, *n*, the GWR equation is described as
$$y_{i}=\beta_{i0}+\overset{p-1}{\underset{k=1}{\sum }}\beta_{ik}x_{ik}+\epsilon_{i}$$
where $y_{i}$ is the dependent variable value at location *i*, $x_{ik}$ is the value of the *k*th covariate at location *i*, $\beta_{i0}$ is the intercept, $\beta_{ik}$ is the regression coefficient for the *k*th covariate, *p* is the number of regression terms, and $\;\epsilon_{i}$ is the random error at location *i*. GWR was widely used in this study to model local relationships between variables, allowing parameters to vary locally.

However, the GWR method results in potential problems in the application. Correlations between GWR estimates exemplify the weakness of GWR [32], indicating that local coefficient maps tend to exhibit multicollinearity and spatial autocorrelation [70]. Given that the multicollinearity and correlation between local coefficients lead to biased estimations and interpretations, GWR is not appropriate for local model specification. To solve multicollinearity problems in the GWR, the lasso regression approach was proposed [24].

The lasso is a shrinkage and variable selection method for regression models that introduces a constraint on the model parameters. Consequently, lasso regression decreases the residual sum of the squares subject to the sum of the absolute value of the coefficients being less than a constant, thereby minimizing the prediction error [71]. In addition, the lasso reduces some coefficients to zero, leading to enhanced model interpretability [71]. The lasso coefficient is defined as
$${\hat{\beta }}^{Lasso}=arg\;\underset{\beta }{\min}\left\{\overset{n}{\underset{i=1}{\sum }}\left(y_{i}\right.-\beta_{0}-\overset{p}{\underset{k=1}{\sum }}x_{ik}{\left.\beta_{k}\right)}^{2}+\lambda \overset{p}{\underset{k=1}{\sum }}\left|\beta_{k}\right|\right\}$$

In the penalty term $\lambda \sum_{k=1}^{p}\left|\beta_{k}\right|$, lambda (λ) controls the shrinkage amount of the regression coefficients.

Based on the lasso method, the GWL is an advanced GWR application that is used to address the multicollinearity problem by adding a lasso constraint in the parameter estimation. In many cases, the GWL overcame the multicollinearity of GWR and showed better performance than GWR [24,72,73,74]. This study aimed to identify local demographic and socio-economic characteristics for SARS-CoV-2 outbreaks by applying the GWL. In this study, a fixed exponential kernel function was used to create the weight matrix of the GWL. This study used the statistical software program R 3.4.2 (R Core Team, Vienna, Austria) for the statistical analysis and the gwrr package to perform the GWL. The map generation and visualization were conducted using ESRI ArcGIS pro (ESRI, Redlands, CA, USA).

## 3. Results and Discussion

Table 2 presents the Poisson regression results and the GWL coefficient estimates. This study used COVID-19 case data as a dependent variable, which followed a Poisson distribution. Given that the Poisson distribution is the basis for analyzing rare events in epidemiological studies [75,76], it is appropriate to apply Poisson regression to investigate the COVID-19 incidence as a rare infectious disease. Hereafter, Poisson regression results are referred to as the global model.

Regarding global models, all variables were statistically significant, except for the young population in the second period. The R-squared value increased throughout the three periods. Moran’s I for the residuals showed minor positive spatial autocorrelations in both the second and third periods. Since the VIF values of each variable were lower than 3.0, multicollinearity did not affect the regression results [77].

Compared to the global models, the GWL results presented diverse local variations that ranged from negative to positive coefficient values (Figure 3, Table 2). The GWL result was based on the entire period’s data. The maps in Figure 3 show the local coefficients for each variable, classifying regions with colors in different intensities. Statistically significant regions are colored in red and blue. Red represents positive estimates; blue represents negative estimates. Furthermore, regions with diagonal lines (no color) indicate that the coefficients were equivalent to 0, implying no statistical significance. The spatial distribution of the local coefficients shows how variables varied across areal units. Hereafter, the GWL results are referred to as the local model.

(1)Young adults (aged 19 to 34 years)

Young adults were among the three most significant population groups for SARS-CoV-2 transmission across the Seoul metropolitan area. SARS-CoV-2 transmission risks depend on young adults’ activities and lifestyles. Regions with a high concentration of young adults need in-depth monitoring of the potential SARS-CoV-2 spread.

Young adults showed a positive coefficient and statistical significance according to the global model (Table 2). This implied that regions with a larger number of young adults are more vulnerable to SARS-CoV-2 transmission. The Itaewon mass infection case of May 2020 exemplified the potential of SARS-CoV-2 spread due to young adults in the first period [35].

The local model confirmed that young adults played a critical role in SARS-CoV-2 transmission in specific regions. First, as a representative region for nightlife in the Seoul metropolitan area, Itaewon is shown in a light red (Figure 4a), implying that young adults acted as SARS-CoV-2 spreaders. All the confirmed cases from the Itaewon mass infection were young adults [35].

Second, regions with commercial districts were related to more SARS-CoV-2 outbreaks. Regions with red in Figure 4b have a higher commercial district ratio than Seoul’s average [78]. COVID-19 risks in commercial districts are higher than in residential areas since commercial districts have larger floating populations compared to residential areas. Floating populations in the commercial districts are mainly composed of young adults.

Third, northern regions with several military bases presented higher SARS-CoV-2 transmission risks. In the red regions (Figure 4c), several military bases with high concentrations of young military members are located along the borderline between South and North Korea. Young Korean males must provide military services by being stationed in military bases for one and a half years. Several mass infection cases were reported within the military bases [79].

Locally targeted interventions for young adults, such as a vaccination certificate, are required. To access crowded public spaces, such as nightclubs and commercial districts, this certificate must be requested. The young adult vaccination rate is the lowest out of all the generations, except for the elderly over 80 years old and minors [80]. Therefore, the national government must aim to decrease this vaccine hesitancy to minimize the local COVID-19 risks that are triggered by young adults. Vaccine injury compensation and vaccine incentive programs for young adults need to be guaranteed.

(2)Christian population

The Christian population was also identified as one of the three most significant population groups for SARS-CoV-2 transmission across the Seoul metropolitan area. Violating public health orders, close contacts during worship affected the SARS-CoV-2 spread [81]. Locally targeted intervention policies for regions with larger Christian populations need more voluntary cooperation from churches.

The Christian population showed a positive coefficient and statistical significance throughout the study period (Table 2). Regions with larger Christian populations were more vulnerable to SARS-CoV-2 outbreaks. Mass infections that were initiated at the Christian churches were frequently observed throughout the study period. In particular, in the first period, mass infections from Christian churches accounted for 22% of total mass infections [82].

The local model presents which specific regions had stronger relationships between the Christian population and COVID-19, showing the mass infection cases. In the first period, the Wangsung Church case exemplified mass infections (Figure 5a). In the second period, the Sarang Jeil Christian Church produced the largest number of confirmed cases (1173 people) as a single church (Figure 5b) [79]. In the same period, the Lord’s Fountain Church reported 16 cases as a COVID-19 cluster in Gochon-eup, where about 30 thousand people live (Figure 5c). In the third period, the Seongsuk Church led to an increase in COVID-19 cases (Figure 5d).

Consequently, locally targeted interventions must mandate Christian churches to follow the capacity limit for worship and provide immunization proof. In addition, local churches, as possible mass infection hotspots, need continuous monitoring. Meanwhile, local governments are recommended to provide churches with technical support for online worship. Worship and subsequent cohesion among communities result in upholding public health norms [83].

(3)Subway commuters

Subway commuters were one of the three most significant SARS-CoV-2 transmission factors in the study. Given that crowded subways increase SARS-CoV-2 infection rates [55], subway commuters in the Seoul metropolitan area were highly susceptible to COVID-19.

In the global models, subway commuters presented a positive relationship between public transport and SARS-CoV-2 transmission. This result implied that regions with more subway commuters were more susceptible to COVID-19 risks. Given that the daily ridership of the Seoul metropolitan area was about 5.5 million in 2020 [84], it was only natural to expect that subways were vectors of SARS-CoV-2 transmission.

The local model confirmed that most regions with subway lines showed positive coefficients and statistical significance (Figure 6). In particular, in suburban regions, subway commuters played a critical role in SARS-CoV-2 transmission. Despite the suburban region’s low population density, many suburban regions showed statistical significance (Figure 6a). This implied that subway commuters from suburban regions spread SARS-CoV-2 across the Seoul metropolitan area.

In the Seoul metropolitan area, around 16.3% of total commuters use the subway every day [85]. This accounts for the highest proportion (31.6%) out of all the public transportation for commuting modes, such as buses, trains, bikes, and taxis [85]. This implies that 1.7 million people are potentially vulnerable to SARS-CoV-2 infections. In addition, the average travel time for subway commuters in the Seoul metropolitan area is 57 min, the longest among the 23 OECD countries [86]. In this situation, the government must monitor SARS-CoV-2 transmission in subway commutes.

However, despite the high transmission risk, COVID-19 cases from the subway have not been officially reported. Public transportation is not subject to epidemiological investigations in Korea. Specific COVID-19 cases from subways are technically invisible. Therefore, preventive interventions for subway commuters must be emphasized more. During rush hour, for example, reducing the interval between subway trains would help to mitigate the concentration of commuters, leading to a decrease in SARS-CoV-2 transmission risks.

(4)Socially vulnerable group variables: older population (aged 65 years and above), foreign-born population, and low-income households

The three variables (older population, foreign-born population, and low-income households) reflected the social vulnerability of the population. These socially vulnerable populations are susceptible to SARS-CoV-2 infection risks [87]. Global models identify relationships between COVID-19 and the socially vulnerable group variables.

In the global models, the older population showed a negatively significant relationship with COVID-19 throughout the study period (Table 2). Regions with fewer elderly were likely to have higher SARS-CoV-2 infection risks. In the Seoul metropolitan area, the older population tends to live in suburban regions, where a lower floating population and fewer commercial stores exist [88].

The foreign-born population presented inconsistent relationships with COVID-19 (Table 2). Regions with higher foreign populations were more susceptible to SARS-CoV-2 infections in the first and third periods. In contrast, the second period showed a negative relationship.

Low-income households showed a positive coefficient and statistical significance throughout the study period (Table 2). Regions with a higher proportion of low-income households were more susceptible to COVID-19 risks.

Figure 7 presents the local model results specifying regions with high COVID-19 risks that were related to socially vulnerable populations (older population, foreign-born population, and low-income households). This map was generated by overlapping the GWL coefficients of the social vulnerability group variables. The colored regions (Figure 7a–c) present the positive coefficients and statistical significance in one or two variables. Many suburban regions showed clusters of socially vulnerable populations and the corresponding high COVID-19 risks, necessitating spatially targeted interventions that are tailored to the local characteristics.

One cluster found in Pyeongtaek-si requires dedicated local interventions for migrant workers’ health (Figure 7a). Including Pyeongtaek-si, regions with green outlines are characterized by the concentration of migrant workers and manufacturing industries [65,89]. Given that migrant workers are easily isolated from COVID-19 prevention guidelines [90], locally targeted interventions for migrant workers must be intensified. In particular, the local interventions need to ensure the safety of housing and working environments, as well as equitable healthcare access [91]. Further, intervention policies to prevent stigmatization and discrimination against migrant workers must be carefully executed to mitigate the negative impact of COVID-19 [92].

Gwangju-si needs interventions that focus on medical resources support (Figure 7b). Gwangju-si has the lowest number of medical doctors per 1000 people in the Seoul metropolitan area [93], thereby lacking the medical capacity for COVID-19 treatment. Since the scarcity of medical resources decreases COVID-19 prevention [94], proactive medical support in Gwanju-si is necessary to control the transmission. In particular, intervention policies in Gwangju-si need to expand healthcare worker availability.

Gimpo-si needs targeted interventions to control subway commuters (Figure 7c). Adjacent to Seoul, 30% of Gimpo-si commuters travel to and from Seoul [95]. In particular, the subway overcapacity during rush hour is notoriously well-known [96], which leads to close personal contact and even possible SARS-CoV-2 infection. Therefore, intervention policies in Gimpo-si must focus on subway control by increasing the number of subway trains and reducing the subway interval time.

## 4. Conclusions

This study investigated the local demographic and socio-economic characteristics of SARS-CoV-2 outbreaks in the Seoul metropolitan area. The GWL identified statistically significant local characteristics that contributed to the SARS-CoV-2 transmissions for each region. This local model showed region-specific COVID-19 cases and specified regions that need locally targeted interventions. Locally targeted intervention policies the apply the GWL at the regional level are capable of controlling SARS-CoV-2 transmission.

The local model results showed that young adults, the Christian population, and subway commuters played critical roles in the SARS-CoV-2 transmission across the Seoul metropolitan area. Notably, close contacts that take place in young adults’ lifestyles, Christian worship, and commutes using public transportations produced SARS-CoV-2 transmission and infections. Regions with larger population groups of these variables were more susceptible to COVID-19 risks.

Locally targeted interventions for young adults need in-depth monitoring for the regions with a high concentration of young adults. Particularly for nightclub and commercial district areas, local interventions must request vaccination certificates for public accessibility. For any violation, heavy fines and strong penalties, including the closure of business, must be imposed. Meanwhile, the national government must guarantee vaccine injury compensation and incentive programs to mitigate vaccine hesitancy.

Local interventions for the Christian population need close cooperation with Christian churches. Under the COVID-19 crisis, capacity limits and vaccinations for worship must be mandatory. Meanwhile, given the social impact of the church regarding upholding public health norms, local governments should provide churches with technical support to boost online worship.

Concerning local transmission via subway commuters, preventive intervention to decrease the crowdedness in subway trains must be introduced, such as reducing the interval between subway trains during rush hour. Further, COVID-19 epidemiological investigations should be extended to at least the subway station to investigate transmission risks via subways.

The local model suggested specific regions that require dedicated care for socially vulnerable population groups (the older population, foreign-born population, and low-income households). These regions are concentrated outside Seoul. In particular, Pyeongtaek-si, Gwangju-si, and Gimpo-si require regionally specified intervention policies that are tailored to local characteristics. Each region needs targeted interventions for the COVID-19 burden on migrant workers, medical scarcity, and subway overcapacity.

This study had limitations. First, this study used aggregated data by administrative level, not the lowest level. Local models at a finer scale can explain more detailed local characteristics that are related to SARS-CoV-2 transmission and pinpoint more specified regions that need targeted interventions. Second, this study did not use other local characteristics that were related to SARS-CoV-2 transmission, such as regional health conditions, educational level, and unemployment.

## Figures and Tables

**Figure 1 ijerph-18-12595-f001:**
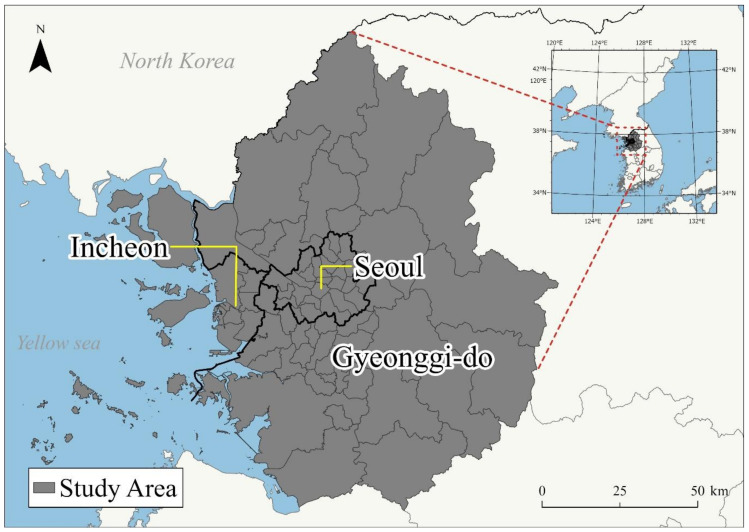
The study area of the research.

**Figure 2 ijerph-18-12595-f002:**
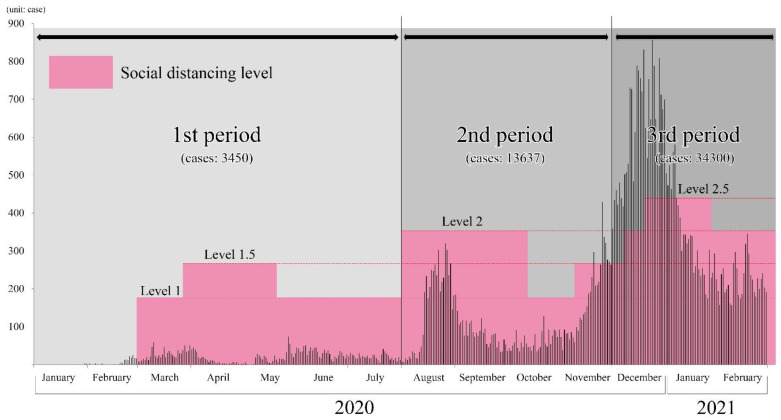
COVID-19 cases and intervention policy levels during the study period.

**Figure 3 ijerph-18-12595-f003:**
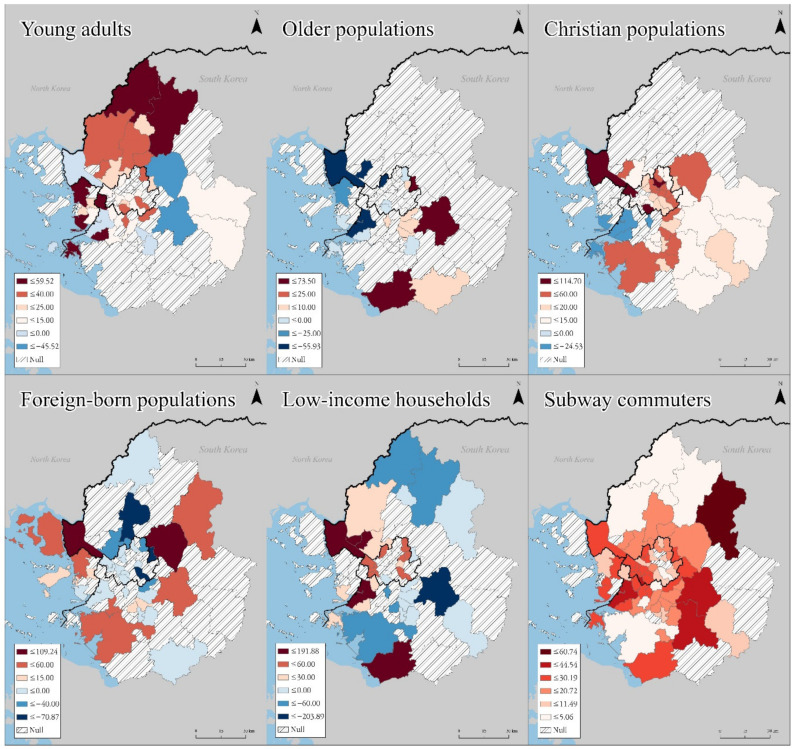
Local coefficient maps for each variable in the GWL.

**Figure 4 ijerph-18-12595-f004:**
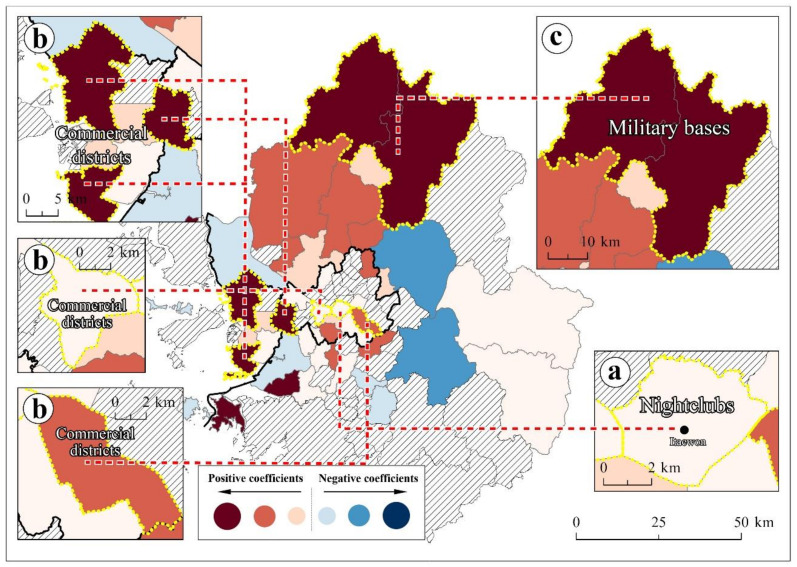
Local coefficients of the young adults variable ((**a**): Itaewon, (**b**): commercial districts, and (**c**): military bases).

**Figure 5 ijerph-18-12595-f005:**
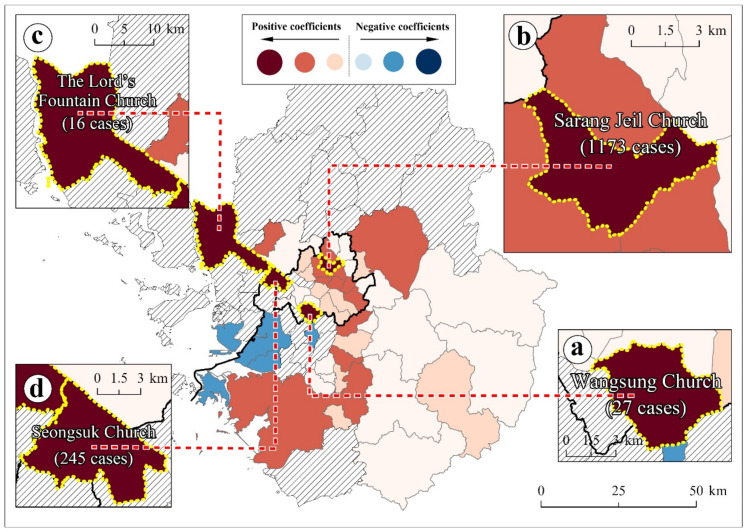
Local coefficients of the Christian population variable ((**a**): Wangsung Church, (**b**): Sarang Sarang Jeil Church, (**c**): The Lord’s Fountain Church, and (**d**): Seongsuk Church).

**Figure 6 ijerph-18-12595-f006:**
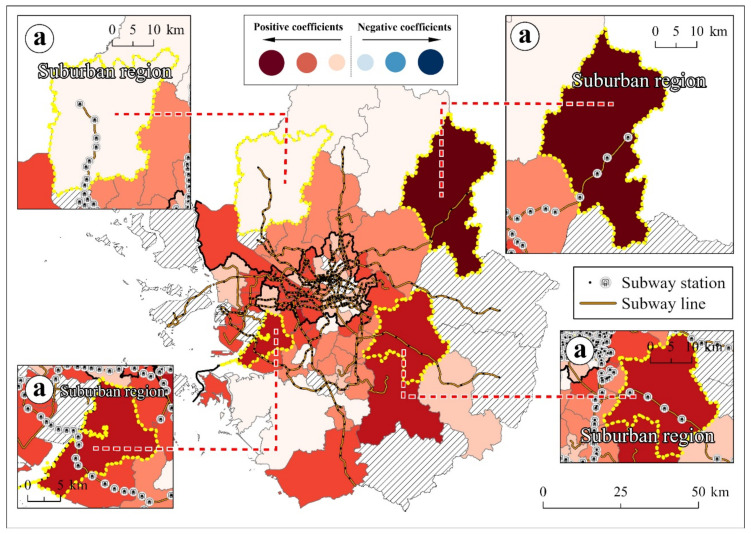
Local coefficients of subway commuters variable ((**a**): suburban regions).

**Figure 7 ijerph-18-12595-f007:**
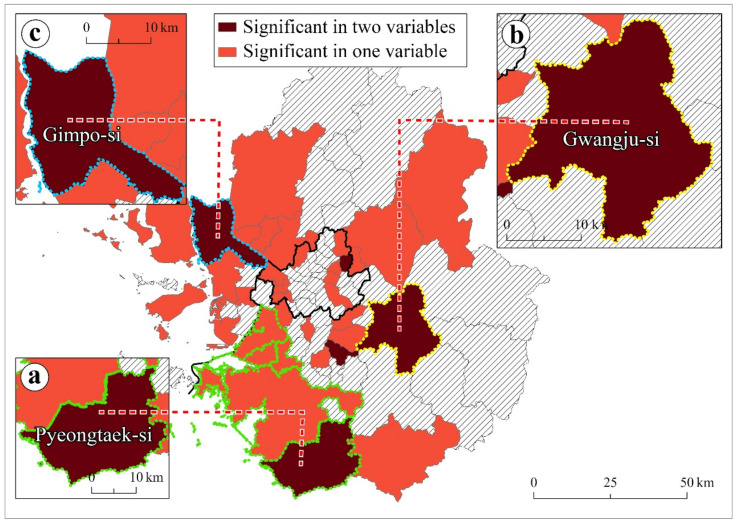
Regions of significant coefficient variables related to COVID-19 risks among the socially vulnerable groups ((**a**): Pyeongtaek-si, (**b**): Gwanju-si, and (**c**): Gimpo-si).

**Table 1 ijerph-18-12595-t001:** Descriptive statistics and references for study variables.

Categories	Variables	Unit	Mean	References	Source
Dependent variable	COVID-19 cases	Case	667.36	-	Seoul, https://data.seoul.go.kr, Gyeonggi-do, https://www.gidcc.or.kr, Incheon, https://www.incheon.go.kr
Independent variables	Young adults (aged 19 to 34 years)	%	20.47	[35,36,37]	KOSIS, http://www.kosis.kr
	Older populations (aged 65 years and above)	%	15.50	[38,39,40,41]	KOSIS, http://www.kosis.kr
	Christian population	%	23.24	[42,43,44,45]	KOSIS, http://www.kosis.kr
	Foreign-born population	%	3.17	[46,47,48,49]	KOSIS, http://www.kosis.kr
	Low-income households	%	3.74	[50,51,52,53,54]	KOSIS, http://www.kosis.kr
	Subway commuters	%	12.48	[55,56,57,58,59,60]	KOSIS, http://www.kosis.kr

**Table 2 ijerph-18-12595-t002:** Summary of the Poisson regression results and the GWL coefficient estimates.

Poisson Regression Results (Global Model)								
Variables	First Period (January~July 2020)		Second Period (August~November 2020)		Third Period (December 2020~February 2021)		Entire Period (January~February 2021)	
	Estimate	*p*-Value	Estimate	*p*-Value	Estimate	*p*-Value	Estimate	*p*-Value
Young adults (aged 19 to 34 years)	2.16 *	0.01	0.43	0.33	1.10 **	<0.01	0.73 **	<0.01
Older population (aged 65 years and above)	−10.87 **	<0.01	−6.26 **	<0.01	−8.63 **	<0.01	−8.11 **	<0.01
Christian population	3.70 **	<0.01	3.43 **	<0.01	3.35 **	<0.01	3.43 **	<0.01
Foreign-born population	1.83 *	0.01	−3.31 **	<0.01	1.96 **	<0.01	0.68 **	<0.01
Low-income households	3.32 *	0.04	3.35 **	<0.01	3.85 **	<0.01	3.72 **	<0.01
Subway commuters	3.62 **	<0.01	3.73 **	<0.01	4.09 **	<0.01	3.96 **	<0.01
R^2^	0.3646		0.4122		0.4957		0.4848	
AIC	1718.35		4083.27		7069.57		10,561.96	
Moran’s I of the residuals	0.03	0.11	0.13	0.02	0.14	0.02	0.11	0.02
**GWL Coefficient Estimates (Local Model)**								
Variables			Max		Mean		Min	
Young adults (aged 19 to 34 years)			59.52		14.89		−76.68	
Christian population			114.70		14.28		−64.62	
Older population (aged 65 years and above)			73.50		−8.21		−101.59	
Foreign-born population			109.24		−5.01		−80.91	
Low-income households			191.88		8.43		−203.89	
Subway commuters			60.74		18.81		1.08	

* *p*-value < 0.05; ** *p*-value < 0.01; AIC: Akaike information criterion.

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
