# Peer review of "Local Characteristics Related to SARS-CoV-2 Transmissions in the Seoul Metropolitan Area, South Korea"

_ijerph, 2021, doi:10.3390/ijerph182312595_

Round 1

Reviewer 1 Report

The article deals with important problems referring to COVID-19 pandemics, such as transmission of the virus among selected social groups and present really interesting research results. However, proposed publication need to be improved on the field of:

  1. theoretical background referring to the most important research on transmission of the virus
  2. discussion on presented results
  3. more in-depth conclusions

Reviewer 2 Report

This paper investigates the characteristics of COVID-19 outbreaks to provide locally targeted intervention policies. Researches about this topic are of practical significance. So far, many articles have analyzed the factors and characteristics of COVID-19 outbreaks. However, this paper doesn't point out the between itself and the previous articles.The innovation of this paper is not obvious.

The main research method used in this paper is Geographically Weighted Lasso (GWL), but there is a lack of analysis of the applicability of the method to the research data. Why this method is better than others when analyzing the characteristics of COVID-19 outbreaks? In addition, I think the technical route of the analysis is not clear and it is better to describe it as a whole in the method part.

The purpose of this paper is to help to advance intervetion policies. But I don't think the results of the paper have more contributions than the existing findings in the field of epidemiology. To government policies,

the conclusions drawn from the research results are too general and lack clear direction and high enforceability.

Finally,I think the following specific shortcomings must be fixed:

Line 19、98:"1st period (from January 2020 to July 2021)" is inconsistent with Figure 2. Or it may be "1st period (from January 2020 to July 2020)"?

Figure 2: What does the Y-axis mean? The axis title is necessary. What does the "level" mean? This should be explained clearly.

Reviewer 3 Report

Thank you for the opportunity to review the interesting article.

Concerning ethical issues you will be careful in migrant workers - foreign-born population (In some case they are living for 10- 20 years in Soul with foreign birthplace? Is it correct?) and Christian population. Do you have the full description of the population about their religion of people? (for comparisons with other groups of religion).

How important is subway commuting? What about other ways of commuting-transport. I have no imagination about the portion of commuters type of transport - railway, bus, tram. The subway is more important. Please, add any general description about the type and amount of passengers by the types of transport in the Soul area.

Figure 4. There is a wrong description of 4a in text like light red color. But 4a extract and also in the map, there is white color. Please, correct this discrepancy.  The same 4b extract- white color. Moreover, the question about the location of nights club and accommodation of young adults. I suppose they are not the same, I suppose that they are different. The nightclubs were reported by young people as the source of infection?

Did you assume the Simpson effect (Simpson's paradox from statistics)? Imagine: when in some districts live more older people than others, it is evident that the count of older COVID people must be higher than other counts of COVI|D people in other groups. The portion must be considered. Please, discuss it in methodology and take it in mind.

Typesetting:

row 190 - correctly is ArcGIS Pro

Round 2

Reviewer 1 Report

The introduced changes allow for the acceptance of the article in the presented form

Author Response

Thank you. 

We checked spelling mistakes.

Reviewer 2 Report

I’m glad to see that the method part has been complemented to make it clearer. However, I believe the paper needs some refinements before it is in good shape to be published. The paper is lacking sufficient citations to relevant studies in analyzing the factors and characteristics of COVID-19 transmission, such as Andersen et al. (2021), Ehlert (2021), Maiti et al. (2021), Xie et al. (2020) and others. How does your approach compare to these other studies? The paper has only discussed the methodological advantages over GWR, but I think other methods which have been applied to this research topic should also be discussed to interpret the innovation of the article compared with previous articles. 

References:

Andersen, L. M., Harden, S. R., Sugg, M. M., Runkle, J. D. & Lundquist, T. E.,2021. Analyzing the spatial determinants of local Covid-19 transmission in the United States, Science of The Total Environment. 754(142396.

Ehlert, A.,2021. The socio-economic determinants of COVID-19: A spatial analysis of German county level data, Socio-Economic Planning Sciences., 101083.

Maiti, A., Zhang, Q., Sannigrahi, S., Pramanik, S., Chakraborti, S., Cerda, A. & Pilla, F.,2021. Exploring spatiotemporal effects of the driving factors on COVID-19 incidences in the contiguous United States, Sustainable Cities and Society. 68(102784.

Xie, Z., Qin, Y., Li, Y., Shen, W., Zheng, Z. & Liu, S.,2020. Spatial and temporal differentiation of COVID-19 epidemic spread in mainland China and its influencing factors, Science of The Total Environment. 744(140929.
